



# The presence of clouds lowers climate sensitivity in the MPI-ESM1.2 climate model

Andrea Mosso[1], Thomas Hocking[1], and Thorsten Mauritsen[1]

[1]Department of Meteorology, Stockholm University, Stockholm, Sweden

**Correspondence:** Andrea Mosso (andrea.mosso@misu.su.se)

**Abstract.** Clouds affect the sensitivity of the climate system by changing their distribution, height and optical properties under climate change. Additionally, clouds have a masking effect on $CO_2$ forcing and can affect other feedback mechanisms such as the surface albedo feedback. To shed light on the overall effect of clouds, we compute how much the equilibrium climate sensitivity to a doubling of $CO_2$ (ECS) changes when clouds are made transparent to radiation in an Earth system model (MPI-ESM1.2). Practically, to stabilise the model climate at near pre-industrial temperatures the solar constant was reduced by 8.8 percent. Our experiments suggest that clouds play a stabilising role in the model, with a clear-sky ECS of 4.34 K, which is higher than the corresponding full-sky ECS of 2.80 K. Detailed partial radiative perturbation diagnostics show that clouds strengthen the lapse rate and water vapour feedbacks and dampen the albedo feedback.

## 1  Introduction

The radiative balance at the top of atmosphere (TOA) determines the state of the climate system. When the system is in equilibrium the mean imbalance is zero, but when the system is forced, a positive or negative imbalance can arise, resulting in warming or cooling of the system. The magnitude of climate change is usually quantified as the equilibrium climate sensitivity (ECS), which is defined as the long term temperature change when the forcing is a doubling of the $CO_2$ concentration relative to pre-industrial levels. The ECS assessed value by the IPCC AR6 is 3 K, with a *likely* range from 2.5 K to 4 K (Forster et al., 2021). In recent years, after having been the main tool to assess the ECS for decades, climate model simulations have been largely shelved in favour of other lines of evidence that allowed the ECS uncertainty to be narrowed down by close to a factor of two (Sherwood et al., 2020; Forster et al., 2021). This seemed timely since the Coupled Model Intercomparison Project phase 6 (CMIP6) showed, across 27 GCMs, a wider range of ECS than previous phases. The causes of this widening have been partly traced back to model representation of cloud processes, not all of which are completely understood (Zelinka et al., 2020; Flynn and Mauritsen, 2020).

Generally, ECS can elegantly be estimated, in a linear framework (see §2.1), as the temperature change that would re-establish radiation balance at the TOA, and it can be expressed in terms of a radiative forcing $F$ and a linear response to temperature changes $\lambda$, the feedback parameter (Gregory et al., 2004). The approach that is usually called process understanding builds on this simple formula, separately estimating the forcing $F$ and the feedback $\lambda$ using different lines of evidence. Recent studies have shown how it is possible to derive analytical expressions for the feedback and the forcing in the spectral





space by making only a few reasonable assumptions, mainly on the dependencies of relative humidity on temperature and spectral properties of water vapour and $CO_2$ absorption. For example, the works of Jeevanjee et al. (2021) and Stevens and Kluft (2023) have focused on the calculation of the clear-sky ECS by computing the effects of the atmospheric components on the feedback and the forcing. The effects of water vapour and $CO_2$ are computed one after the other, however, introducing the

role of clouds in this analysis is complicated, given the wide range of mechanisms associated with them. Therefore, Stevens and Kluft (2023) conclude that, within their framework, it is not yet evident if the clouds play more of a stabilising or a destabilising role, since cloud feedback and masking effects on both forcing and feedback push the ECS in different directions. As a consequence, the cloud effect on ECS can be thought of as a correction to a clear-sky ECS, which in turn depends mostly on physical mechanisms that we can quantify with higher confidence than what we can for clouds.

Motivated by these ideas of estimating ECS without clouds influencing the result, we set out to test this idea in a global climate model. We achieved this by making clouds transparent to all radiation and compensating for the warming that would result by adjusting the solar constant. Essentially one can think of the present study as complementary to the one dimensional models proposed earlier, in that we additionally simulate the global circulation response to the presence- (full-sky, FS), and absence of clouds (clear-sky, CS). We furthermore apply the Partial Radiative Perturbation (PRP) technique to separate the

feedbacks and forcing contributions, to elucidate which mechanisms differ between the two simulations.

## 2  Methods

Our analysis was carried out by performing simulations with the Max Planck Institute Earth System Model version 1.2 (MPI-ESM1.2). The next sections describe the linear framework and the general procedure used for feedback and forcing estimations, the peculiarities of the model we used, the setup necessary for the Clear-Sky simulations and the experiment design.

### 2.1  The linear framework and the regression method

To study the temperature response to the forcing, a linear framework (Forster et al., 2021) is used, writing the TOA imbalance $N$ as the sum of an Effective Radiative Forcing $F$ (ERF in the following, Sherwood et al., 2015) and a contribution linearly proportional to the surface air temperature change $\Delta T$:

$$N = F + \lambda \Delta T, \tag{1}$$

where the strength of the effect that the temperature change feeds back to the TOA imbalance is given by the feedback parameter $\lambda$. Typically, it is assumed that the feedback parameter is state and time-independent. This assumption, which is not always perfect (Ceppi and Gregory, 2019) is generally accepted when the magnitude of the forcing is small enough, as is the case with $CO_2$ concentration doubling.

In this framework it is often assumed that different feedback mechanisms act independently from each other, allowing for

a perfect separation of the total imbalance into as many terms as the number of mechanisms affecting it. Traditionally, the





imbalance is separated into contributions from $CO_2$, temperature, water vapour, surface albedo and clouds:

$$N = N_{CO2} + N_T + N_{WV} + N_\alpha + N_C \tag{2}$$

where the temperature contribution is further divided into stratospheric (to separate effects of the well known stratospheric cooling), Planck and lapse rate tropospheric terms. In principle, additional feedback mechanisms may apply for forcings other than $CO_2$ concentration changes, which would result in corresponding contributions to the imbalance. However, for the purpose of this paper, we use the traditional separation.

The well established Gregory regression method (Gregory et al., 2004) is used to compute the ECS and the contributions to the total feedback from different mechanisms (see §2.5), whereby ordinary least squares linear regression being applied to the first 100 years of each simulation, unless otherwise stated. For the ERF contributions we complement the Gregory derived estimates with estimates based on the the fixed-SST method (Hansen et al., 2005). In these experiments the forcing is estimated from a fixed-SST simulation as

$$F_S = N_0 - \lambda \delta T_0 \tag{3}$$

with $N_0$ the average TOA imbalance over 30 simulation years, $\lambda$ the feedback parameter from the regression method, and $\delta T_0$ the 30 years average global mean temperature change given by land temperatures not being fixed. The second term thus accounts for the small temperature response to the forcing, and averaging over 30 years accounts for the internal variability.

## 2.2 Model properties

MPI-ESM1.2, a coupled Earth system model, was used in its lowest resolution, i.e. CR (Mauritsen et al., 2019) to allow us to perform longer simulations and investigate the dependency of our results on slow adjustments and initial conditions. MPI-ESM1.2-LR has an $ECS = 2.77$ K and our computed value for MPI-ESM1.2-CR is $2.80$ K, placing it on the lower half of the AR6 assessed *likely* range of 2.5-4.0 K.

Cloud mechanisms have been identified as one the main sources of the ECS spread among different models (Bony et al., 2006, 2015; Zelinka et al., 2020; Flynn and Mauritsen, 2020), and for this reason it is important to bear in mind how the model we use represents their behaviour under warming. MPI-ESM1.2 shows a positive cloud fast adjustment (Andrews and Forster, 2008; Kamae et al., 2015) and an overall positive cloud feedback (Zelinka and Hartmann, 2010).

Given that both the feedback and the forcing contributions are positive, the clouds in MPI-ESM1.2-CR directly destabilise the climate, by raising its ECS. We can quantify this effect by isolating the cloud contributions and computing the diagnostic Clear-Sky ECS that we would obtain if the effect of making the clouds transparent to radiation were only that of zeroing their feedback ($\lambda_C$) and fast-adjustment ($f_C$) contributions in the Full-Sky experiment:

$$ECS_{CS}^{\perp\!\!\!\perp} = ECS_{FS} \underbrace{\left[ \frac{\lambda_{FS}}{\lambda_{FS} - \lambda_C} \cdot \frac{F_{FS} - f_C}{F_{FS}} \right]}_{\xi^{-1}} \tag{4}$$

The diagnostic estimate of the direct effect that clouds have on the ECS is $\xi = 1.18 \pm 0.04$. This value can be compared with the effect that removing the cloud radiative effects has, which accounts for the masking effects and the interactions between





clouds and other feedback mechanisms. Similarly, introducing $ECS_{CS}^{\parallel}$ in the comparison with $ECS_{FS}$ and $ECS_{CS}$ helps to separate the effect on the ECS that clouds directly have through their feedbacks from the one they produce affecting the others. This could be of great value when comparing models with different cloud feedback strengths.

## 2.3 Cloud transparency and solar constant adjustment

In the clear-sky simulations the clouds are made transparent to radiation of every wavelength. This is achieved by setting the cloud water path equal to zero in the radiation calculations, such that effectively the cloud optical depth is zero. In the standard MPI-ESM1.2 control climate the cloud radiative effect (CRE) results in a strong cooling in the shortwave, as clouds reflect solar radiation, and warming to a less extend in the longwave (Boucher et al., 2013). The net global mean CRE being negative implies that removing it would lead to a much warmer climate, our estimate from Gregory extrapolation being approximately 22 K warmer than the current climate, and hence the results would be less relevant to the goals of the current study. For this reason we need to compensate for this warming, and we achieve it by reducing the solar constant.

The intensity of the reduction is first computed theoretically by matching the extrapolated ERF of the CRE removal with the reduction in the incoming shortwave radiation. Such simulations are nevertheless left with a residual $4.67$ Wm$^{-2}$ ERF, showing the nonlinear effects of two big forcings compensating each other. Consequently, the solar constant is further reduced with a tuning process aiming at getting as close as possible to initial zero imbalance. Our final value for the solar constant multiplication factor is $0.912$, meaning a reduction by $-8.8\%$ of its magnitude. This value is then used for all the clear-sky simulations henceforth.

Given the strong differences in the spatial patterns of the two forcings we applied when compensating the removal of cloud radiative effects with the reduced solar constant, we analyse how letting the system equilibrate on long timescales affects our findings. As shown in §3.1, waiting for 800 years rather than 100 does not yield substantially different results, and only marginally enhances differences that are already discernible after 100 years.

## 2.4 Clear-sky simulation peculiarities

Whilst the ECS is a metric defined using global mean quantities, SST patterns are known to affect climate feedbacks (Andrews et al., 2015; Armour et al., 2013; Ceppi and Gregory, 2017). Hence it is worth describing them in the clear-sky simulation. As the clear-sky system was tuned aiming at zero initial imbalance, we used surface air temperature timeseries of the clear-sky simulation to compare with the full-sky pre-industrial control simulation, which for MPI-ESM1.2-CR was spun up from the ocean state of the previous version and fine tuned until quasi-stationarity was reached (Mauritsen et al., 2019). Temperature variations standing out over the internal variability are observed in the first decades after the tuning, motivating the choice of performing $CO_2$ doubling experiments either after 100 or 800 years.

The clear-sky control simulation undergoes a slight cooling relative to the full-sky simulation, and it is on average $0.61$ K colder after 900-1000 years. Temperature differences are more pronounced over land ($-2.08$ K) whilst mid-latitude oceans are warmer ($+0.59$ K). An increase is found in the Southern Ocean sea ice, by almost $100\%$ in the annual average extent and more than $150\%$ in volume. This is a major cause of the albedo feedback increase observed in the clear-sky experiments (§3.1).





## 2.5 Experiment design and feedback decomposition

Despite the ECS being defined as the temperature difference after a $CO_2$ concentration doubling, the standard experiment, in previous studies and in the CMIP5, was the *abrupt4xCO2*. This was chosen in order to obtain a higher signal-to-noise ratio in the Gregory plot, assuming the forcing from $CO_2$ doubling is half that of $CO_2$ quadrupling. Nevertheless, it turned out that many models show a significant ECS increase with warming because of non-linearity that arises when the forcing of the system is not small enough (Jonko et al., 2013; Meraner et al., 2013). Consequently, we decided to perform the $CO_2$ doubling experiment. As the effect of cloud transparency on feedback mechanisms could not necessarily stand out from the internal variability of the system, we account for the variability by performing an ensemble of 10 experiments (*abrupt2xCO2*) with different initial conditions. This way, differences that stand out from the ensemble spread can be considered significant. The ten ensemble members are simulations starting 20 years apart from each other, running for 100 years.

The feedback decomposition is achieved by making use of an online partial radiative perturbation technique (PRP, Colman and McAvaney, 1997; Wetherald and Manabe, 1988). It is more computationally expensive than the radiative kernel method (Soden et al., 2008), but PRP is far more accurate and it allows for the direct computation of feedback contribution from clouds, which in the kernel method is only computed as a residual (e.g. Soden et al., 2008). The application of the PRP technique with a timestep for the PRP call in the radiation code of 10 hours, resulting in a sampling of the diurnal cycle in five days, shows that the single feedback contributions add up to the total feedback with just a 0.2 percent error.

## 3 Results

By means of Gregory regression over the 10 ensemble members, we estimate the effective ECS of the full-sky and clear-sky experiments in two steps. First, we calculate feedback and forcing for each ensemble member, then, assuming the regressed feedbacks and forcings are taken from a Gaussian distribution whose standard deviation we estimate from the ensemble spread, we determine the ECS distributions. Our results (Fig. 1) show that the clear-sky ECS (4.34 K) is significantly higher than the full-sky (2.80 K), with the clouds having an overall effect opposite in sign to that of just removing their feedback (diagnosed $ECS^{\perp} = 2.36$ K). This suggests that clouds have either a strong radiative effect on other feedback mechanisms or that they mask substantially the forcing from $CO_2$. Which of these effects are at play in MPI-ESM1.2-CR is what we explore in the next sections.

## 3.1 Feedbacks

In the clear-sky simulation the total feedback parameter ($\lambda_{CS} = -0.81 \pm 0.04$ $\mathrm{Wm^{-2}K^{-1}}$) is significantly weaker than in the full-sky simulation ($\lambda_{FS} = -1.30 \pm 0.03$ $\mathrm{Wm^{-2}K^{-1}}$). In Fig. 2 the full-sky and clear-sky individual feedbacks are compared, together with the values obtained by regressing the clear-sky fluxes in the full-sky simulation ($CS_F$). We can consider the latter values as indicative of the immediate impact one would observe by swiftly removing the clouds, without allowing the system to adjust to this change as in the clear-sky simulation (Appendix A). Differences between full-sky and $CS_F$ values



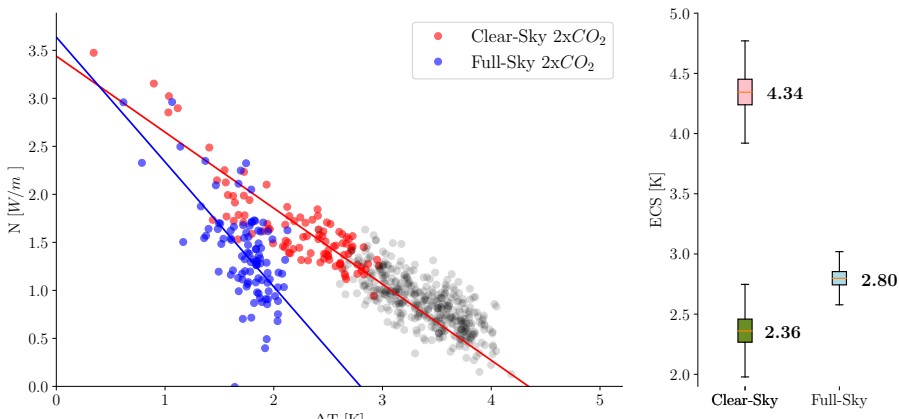

**Figure 1.** Gregory plot for full-sky (blue) and clear-sky (red) experiments. Ensemble averages of the first 100 simulation years are used for the regression. Using 500 simulation years (grey dots), albeit showing a slight bending, does not change the result substantially ($ECS = 4.7$ K). The right panel shows the computed ECS distributions. The green bar plot represents the diagnosed $ECS_{CS}^{\parallel}$ of equation 4.

are representative of the masking effect of clouds, while clear-sky values account for both simple masking and the feedback mechanisms alteration by global circulation response to cloud transparency. The biggest differences between full-sky and clear-sky come from the positive variations in the surface albedo and lapse rate (LR) feedbacks, the latter being only partially compensated by the weakening of the water vapour (WV) feedback.

The weakening of the LR feedback is the most interesting result. It is mostly given by the weakening of the strongly negative LR feedback in the tropical region. The upper tropospheric warming was already found to be cloud-induced (Wetherald and Manabe, 1988; Langen et al., 2012), and Mauritsen et al. (2013) proposed that this could be due to a longwave flux convergence below the cloud tops, as with warming the surface emits more radiation, while anvil clouds emit roughly the same, thus leading to flux convergence and heat accumulation aloft. This might explain the weakening of the LR feedback in the clear-sky simulation.

The WV feedback instead shows a completely different behaviour. Because the water vapour is primarily located at lower altitudes and thus masked by the clouds above it, sudden removal of this masking, as for the $CS_F$ value, results in a stronger WV feedback. Conversely, when the clouds are permanently made transparent to radiation (clear-sky), the weakening of the LR feedback also reduces the WV feedback originating from water vapour at higher altitudes: If there is less warming in the upper troposphere, where humidity can be considered temperature-constant, there would be less high altitude water vapour increase with the same surface warming and hence a weaker WV feedback.

This compensating effect is appreciated only when comparing $CS_F$ and clear-sky since in full-sky clouds mask the two feedbacks differently. The albedo feedback is primarily given by the sea-ice melting in the polar regions. Those regions are generally cloudy, and hence removing cloud radiative effects allows more SW radiation reach the surface, thus enhancing the surface albedo feedback. The agreement between $CS_F$ and clear-sky feedback strength might lead us to think that the effect of





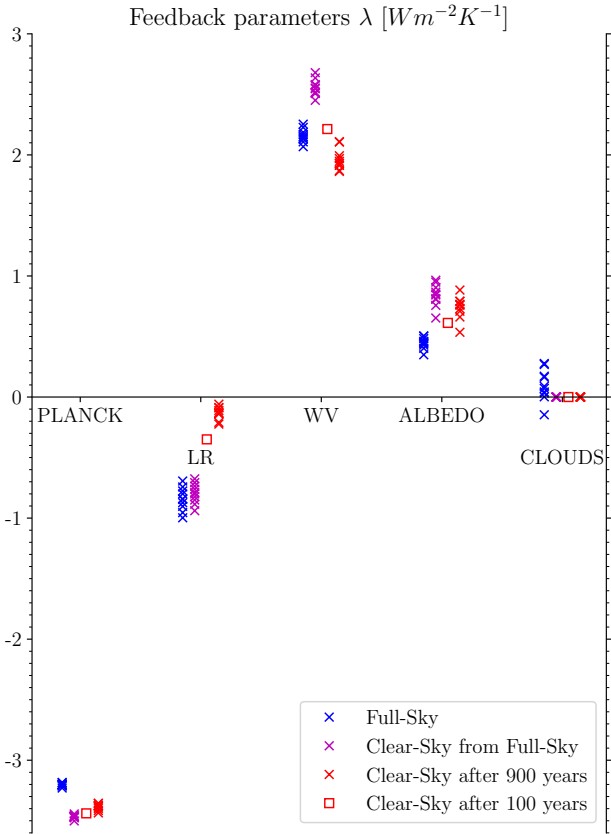

**Figure 2.** Feedback parameters across 10 ensemble members, diagnosed with the PRP technique for the full-sky experiment (blue), clear-sky values from the full-sky simulation (CS$_F$, purple) and the clear-sky simulation (red). The biggest differences between full-sky and clear-sky simulations are in the lapse rate and albedo feedbacks.

clouds is that of pure masking. However, this agreement is purely coincidental, as the growth of new Antarctic sea ice in the clear-sky experiment is responsible for a widening of the area where a positive albedo feedback acts, rather than an increase in feedback intensity in the same regions as would be expected from masking (see Fig. 5).

The Planck feedback shows minor differences, but it is slightly stronger under clear-sky conditions, possibly because the
175    emission level is closer to the surface and thus warmer when clouds are transparent.

## 3.2    Effective radiative forcing

The differences in the ERF between full-sky and clear-sky simulations can not only be traced back to the direct $CO_2$ forcing, but are also related primarily to stratospheric temperature- and cloud adjustments which act to enhance the forcing. When separating the total imbalance into the single components, extra care should be taken in the interpretation of the fast adjustments.
180    When analysing the single contributions, for instance deviations from linearity can be more evident, leading to what we would



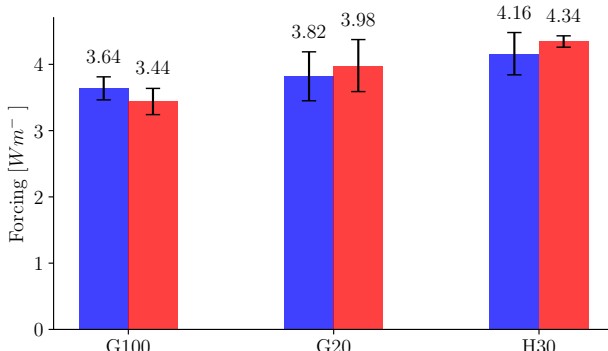

**Figure 3.** Total forcings from full-sky experiment (blue) and clear-sky experiment (red), computed with three different methods. Errorbars are plus and minus one standard deviation of the ensemble spread for *[G100]* and *[G20]*, and likewise of the yearly mean imbalances for *[H30]*.

think are fictitious adjustments because the regression method is extrapolating back to zero temperature change (the y-axis intercept). To take this into account, for our analysis we used three methods to assess the ERF. Here we highlight strengths and weaknesses of each of them:

1. Gregory regression across the entire 100 years of simulation *[G100]*: being the same method as the one used to assess the feedback parameter, it allows to explain how much the contributions to the ERF in the full-sky and in the clear-sky simulation lead to a different ECS. On the other hand, being determined by interpolation of points which get denser further from the y-axis, it leads to a low bias if the regression line bends upwards.

2. Gregory regression of the first 20 years of the simulation, as in Block and Mauritsen (2013) *[G20]*. Using points on a smaller temperature range closer to $\Delta T = 0$, where they are more homogeneously distributed reduces the effect of bending.

3. Fixed-SST 30 years experiments, following the Hansen et al. (2005) method *[H30]*. An advantage of this method is that a fixed SST results in most feedback mechanisms are effectively disabled. Nevertheless, land temperatures may change, and in particular feedbacks occurring over land might be different in the full-sky and clear-sky simulations.

Consequently, when focusing on their impact on the ECS *[G100]* is preferable, while physical interpretation is more straightforward using *[G20]* and *[H30]*.

Results for the total forcing with the three methods are displayed in Fig. 3. Somewhat surprisingly, the enhancement that we expect from removing cloud radiative effects on the forcing in the clear-sky simulation is only observed when using *[G20]* and *[H30]*, though the differences are not statistically significant. Therefore, since there are mechanisms that we expect to be



largely affected by the cloud transparency, it is better to analyse them by separating the total ERF in the direct $CO_2$ forcing and the fast adjustments by means of PRP, as shown in Fig. 4.

In the clear-sky simulation the $CO_2$ direct forcing is enhanced with respect to the full-sky simulation. The three methods show a good agreement, with clear-sky forcings always stronger than full-sky. The effect of clouds damping this forcing, which has long been known (Myhre et al., 1998), is well explained by Stevens and Kluft (2023): Since the temperature of high clouds controls the emission over the $CO_2$ below them, they reduce the effects of changes in $CO_2$ concentration. Nevertheless, the presence of clouds also permits a cloud-adjustment, that is, a change in cloudiness directly in response to the change in $CO_2$ concentration. In MPI-ESM1.2 there is a positive cloud fast adjustment (Fig. 4), primarily due to a reduction in the low-level stratiform clouds in the subtropics. Similar mechanisms have been found in other models (Kamae et al., 2015; Andrews et al., 2012). The removal of this cloud adjustment, though, only partially counterbalances the direct $CO_2$ forcing, such that other small changes to adjustments add up to the nearly identical ERF between the full-sky and clear-sky experiments (Fig. 3).

Differences in the diagnosed adjustments between the three different methods, *[G100, G20, H30]*, can also be artifacts of the bending, or non-linearity, in the Gregory diagram (Block and Mauritsen, 2013). Such behavior is particularly evident for the lapse-rate and water vapour feedbacks, which nearly cancel, as well as for the surface albedo feedback. The artifacts are larger in the clear-sky experiments. The G100 method is mostly affected by the bending, and in most cases the H30 method is least affected. Somewhat surprisingly, the water vapour adjustment computed with *[H30]* is negative. This is due to the Hansen forcing definition, and we explain it in §3.3, as the differences in the spatial distributions are revealing.

### 3.3 Spatial patterns of feedback and forcing

Following Hedemann et al. (2022) we used the global mean temperature definition of local feedback to represent the spatial differences between full-sky and clear-sky simulations. In this way, local feedbacks are linearly additive and local differences can explain global feedback variations. Local feedback differences for each component are plotted in Fig. 5. Two details can be observed from the map of total feedback differences. First, the spatial pattern is strongly influenced by that of the cloud feedback, which is zero in clear-sky. Second, the strongest differences are in the Southern Ocean, where cloud transparency strongly enhances the albedo feedback. The different behavior of Southern Ocean sea ice, which is subject to larger seasonal changes in the clear-sky, also affects the temperature feedback. This feedback is generally stronger at high latitudes under clear-sky. The removal of clouds also plays an important role in damping the positive water vapour feedback at low latitudes, reducing latitudinal variations.

Spatial forcing contributions are shown in Fig. 6. As with the case of feedbacks, the rich spatial pattern observed in full-sky is primarily attributed to the contribution of clouds, which is positive over land and predominantly negative over the oceans. Contributions from other mechanisms exhibit greater spatial homogeneity. It is interesting to observe how the $CO_2$ forcing is influenced by high and thick clouds, with the forcing being more strongly damped in the regions of the Southern Ocean storm tracks and the Asian monsoon.

With respect to the water vapour feedback, it can be puzzling to see a negative global mean adjustment in the fixed-SST scenario. This is an artifact of the local application of equation 3. The term $F_0$ is positive over all land and nearly zero over



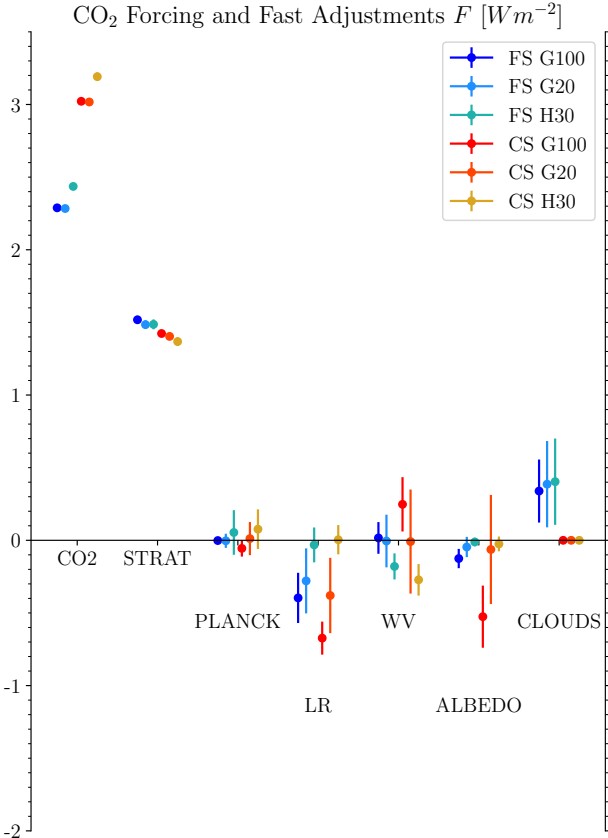

**Figure 4.** $CO_2$ direct forcing and fast adjustments assessed with the PRP technique for full-sky experiment (shades of blue) and clear-sky simulation (red to yellow) using the three methods described in §3.2. . Errorbars are $1\sigma$ of the ensemble spread for *[G100]* and *[G20]*, $1\sigma$ of the yearly mean imbalances for *[H30]*.

the oceans (not shown), while the WV feedback is positive and stronger over the oceans (Fig. 5). This means that removing it from the oceans (where temperatures are held fixed and feedbacks are disabled) introduces a strong negative offset which adds to the less intense removal over lands, resulting in a negative $F_S$.

## 4 Conclusions

By comparing climate change simulations with abruptly doubled atmospheric $CO_2$ using the MPI-ESM1.2 global climate
model with and without cloud radiative effects, we have shown that clouds can exert a stabilizing influence on the climate system. When clouds are made transparent to radiation in the model the equilibrium climate sensitivity increases from 2.80 K to 4.34 K. Whereas the effective radiative forcing is surprisingly close in magnitude, the total negative feedback is substantially weakened when clouds are transparent.







**Figure 5.** Local feedback parameters as averages over the 10 ensemble members, and their differences between full-sky and clear-sky.

In contrast, the MPI-ESM1.2 model's diagnosed cloud feedback and the cloud adjustment to increasing $CO_2$ are both posi-

tive, so should act to enhance global warming. According to these diagnostics, the equilibrium climate sensitivity should have

decreased to 2.36 K when clouds are made transparent. The reason is that clouds have a dual effect in the model. In addition



**Figure 6.** Local ERF contributions and their differences between full-sky and clear-sky, calculated using *[H30]*.

to the positive cloud feedback and fast adjustment to $CO_2$, clouds also mask part of the direct $CO_2$ forcing and they alter the strength of other feedback mechanisms, either through simple masking or by altering the mechanism itself.



When it comes to effective radiative forcing the situation is fairly simple. The masking effect of clouds is to dampen the
direct radiative forcing from a doubling of $CO_2$ by about 0.7 $\mathrm{Wm^{-2}}$. Popularly speaking high clouds mask the $CO_2$ increase
occurring below them. In MPI-ESM1.2 the clouds respond directly to a $CO_2$ increase with a decrease in cloudiness, thereby
increasing the forcing by about 0.4 $\mathrm{Wm^{-2}}$. Together with a number of other small contributions the net effect of clouds on the
effective radiative forcing in this particular model is close to zero.

The situation is more complex when it comes to the effect of clouds on the total feedback. When including clouds the total
feedback is reduced by nearly $-0.5$ $\mathrm{Wm^{-2}K^{-1}}$, despite the fact that the MPI-ESM1.2 model has a positive cloud feedback of
about 0.1 $\mathrm{Wm^{-2}K^{-1}}$.

A naive expectation might be that removing clouds should unmask a stronger surface albedo feedback. This simple idea is
supported by the fact that the diagnosed clear sky surface albedo feedback in the full sky experiment is close in magnitude to
that in the clear sky experiment. However, upon a closer inspection of the spatial distribution of the surface albedo feedback,
this appears to be coincidental since there is a change in the base climate with less snow and ice in the Northern Hemisphere,
and much more abundant sea ice in the Southern Ocean in the clear sky experiment. The end result in both these cases is that
the presence of clouds dampens the surface albedo feedback by about a factor of two, or about $-0.3$ $\mathrm{Wm^{-2}K^{-1}}$.

The largest single contribution to the change in the total feedback comes from the lapse rate feedback, about $-0.7$ $\mathrm{Wm^{-2}K^{-1}}$.
This is partially compensated for by shifts in the water vapour and Planck feedbacks, and if we add up all three feedbacks the
difference is still close to $-0.3$ $\mathrm{Wm^{-2}K^{-1}}$. The much stronger lapse rate feedback is primarily associated with cloud-induced
heating in the upper tropical troposphere, and represents an effect that is not simply a cloud masking effect.

All in all, it should be remembered that the effect of clouds will be model dependent. But also therefore, it is to be expected
that different models without cloud radiative effects will have a response that is more similar to each other than it is with
clouds. In the present study we propose a fairly simple protocol of reducing the solar constant to stabilise the climate near the
pre-industrial state and so it could be easily replicated by other modelling groups.

*Code and data availability.* The source code for MPI-ESM1.2 is available through https://mpimet.mpg.de/en/science/models/mpi-esm (Mauritsen et al., 2019). Model outputs and scripts used to produce the figures for this paper are available at https://doi.org/10.5281/zenodo.10697650
(Mosso et al., 2024).

## Appendix A: Diagnosing clear-sky ECS from the full-sky simulation

What is the meaning of the clear-sky feedback and forcings that we obtain from the clear-sky fluxes in the full-sky simulation
($\mathrm{CS}_F$)? The single calculation that the radiation code performs is exactly the same as the one performed in the clear-sky
simulation, except for the fact that clear-sky values are not used to update physical quantities outside the radiation code in the
full-sky simulation. This means that we can regard the $\mathrm{CS}_F$ as an instantaneous "transient" value between the full-sky and the
clear-sky simulation, which accounts only for the cloud masking. In the clear-sky experiment instead, the system dynamically



reacts to the absence of clouds. It is important to note that, from the clear-sky fluxes within the full-sky simulation, one might be tempted to try to infer a clear-sky ECS. The problem is that whilst the Gregory method builds on the physical argument that the TOA imbalance goes to zero at the equilibrium, such an argument does not hold for the clear-sky fluxes, since the clouds' contributions can change for different equilibrium states and the two contributors to the equilibrium imbalance can change:

$$\Delta N_{eq} = N_{CS} + N_{CL} = 0 \tag{A1}$$

namely $R_{CS}$ and $R_{CL}$ being different for different equilibrium states (with higher $CO_2$ concentrations the low clouds' radiative effect would be partly damped by the extra $CO_2$ above them), where $N = R - R_0$ is the TOA imbalance and $R_0$ the TOA net flux of a reference equilibrium state. In other words, the decomposition of equation A1 is an alternative to that of equation 2, with $N_{CL}$ being the sum of the direct clouds' contribution $N_C$ and their masking effect on all other feedbacks.

*Author contributions.* AM conducted the simulations and analysis and wrote most of the manuscript. All authors contributed to the research and writing.

*Competing interests.* The authors declare no conflict of interest.

*Acknowledgements.* AM thanks Martin Renoult and Linnea Huusko for their assistance in conducting the simulations. Computations and data handling were enabled by resources provided by the National Academic Infrastructure for Supercomputing in Sweden (NAISS) at Stockholm University, partially funded by the Swedish Research Council through grant agreement no. 2022-06725. This project was funded by the European Research Council (ERC) (grant agreement no. 770765), the European Union's Horizon 2020 research and innovation program (grant agreement nos. 820829 and 101003470), and the Swedish Research Council (VR) (grant agreement no. 2022-03262).



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
