# Peer review of "The presence of clouds lowers climate sensitivity in the MPI-ESM1.2 climate model"

_EGUsphere, 2024_

## Referee Comment (RC1)

**Review for manuscript EGUsphere-2024-618– The presence of clouds lowers climate sensitivity in the MPI-ESM1.2 climate model**

**General evaluation: Major revision.**

This manuscript presents a "clear-sky" modeling approach (i.e., making clouds transparent to radiation) for a more complete evaluation of clouds impacts on the equilibrium climate sensitivity (ECS). In complement to the common approach of diagnosing the cloud radiative effects from the full-sky model experiment (both the cloud direct radiative effects and the cloud masking on CO2 and water vapor radiative effects), this clear-sky approach additionally accounts for the cloud indirect effects on thermodynamics (e.g., temperature lapse rate and water vapor distribution) and dynamics (e.g., changes in circulation and the associated surface warming pattern), which are shown to significantly modulate the radiative forcing, adjustments, and feedbacks. As the result, the ECS in the clear-sky experiment is much higher than the full-sky experiments with the same climate model, which is surprising but interesting. The authors further perform a systematic decomposition of the effective radiative forcing (ERF) and feedbacks and show their spatial patterns, which helps clarify the physical mechanisms.

Overall, I think this clear-sky approach is useful in identifying the indirect cloud effects on ERF and feedbacks, which have not been emphasized much in previous studies. Unfortunately, the comparison is heavily impacted by the control climate differences between the clear-sky and full-sky run, which complicates the physical interpretation. Although the authors have tuned the global-mean temperature to be comparable by reducing insolation in the clear-sky run, the temperature spatial pattern (pole-to-equator gradient, land-sea contrast, interhemispheric contrast, etc.) is still drastically different, and many of the adjustments and feedbacks are dependent on these patterns (especially the Antarctic sea-ice cover and its associated albedo feedback). Thus, I think it is a bit far-fetched to attribute the clear-sky versus all-sky ERF and feedback differences simply to the cloud modulation of radiative adjustments and feedbacks. In the current manuscript, this complication by control climate has not been clearly presented or sufficiently discussed.

To address this issue, I think the difference between the control climate states in the clear-sky and all-sky runs should be presented and discussed in further detail. In the current manuscript, it is only mentioned briefly on L116–119 without any figures. I think spatial maps of the top-of-atmospheric (TOA) radiative imbalance and surface temperature for both clear-sky and all-sky runs and their differences should be shown. They are essential for the readers to evaluate the mean-state impacts on ERF and feedbacks (e.g., by comparing them to Figures 5 and 6). In addition, the sea surface temperature (SST) warming pattern (evaluated with the same ensemble members and time spans as Figure 5) should also be presented, so that the readers can link the feedback differences to the known mechanisms of SST pattern effects, and possibly associate them with the SST Green's function studies.

In the result discussions, it would be necessary to distinguish the mean-state induced differences from the cloud-induced differences more clearly. Especially, the impact of the larger

Antarctic ice cover in the clear-sky control climate should be more thoroughly analyzed – it not only leads to a stronger ice albedo feedback, but it is also associated with a southern ocean amplified (and tropically damped) SST warming pattern (implied from Figure 5, "planck" panel), which is known for increasing the climate sensitivity (most recently by Kang, Ceppi, Yu and Kang, 2023, Nature Geoscience). Even in clear-sky experiments without cloud feedbacks, the lapse rate feedback is still impacted by the SST pattern (e.g., Andrews and Webb, 2018, Journal of Climate). This mechanism is implied in Figure 5: the SST over the tropical convective regions in the clear-sky run does not warm as much as the full-sky run, which likely leads to the reduced free-tropospheric warming and lapse-rate feedback globally (compare the "planck" and "lapse-rate" panels). Thus, I think the authors' explanation of the lapse rate feedback differences by the direct cloud radiative heating (L155–160 and L265–266) is likely insufficient.

Additionally, I think it would be necessary to discuss how this clear-sky approach is compared with the cloud-locking approach (e.g., Medeiros, Clement, Benedict, and Zhang, 2021, npj Climate and Atmospheric Science), which can also disable all cloud-induced thermodynamic and dynamic changes in the warming climate but largely maintains the control climate state. Are there significant weaknesses of the cloud-locking approach that motivate the authors to use the clear-sky approach instead? If so, it would be helpful to discuss in the introduction and conclusion sections. If not, the different physical interpretations of these two approaches should still be discussed. It would be even better if the authors can perform cloud-locking experiments and see which of the cloud impacts on ERF and feedbacks are robust for both approaches.

**Specific comments**
- L13: It should be clarified that the ECS is defined by the global-mean surface temperature (GMST).
- L25: Please include the references for the "recent studies" at the beginning of this line.
- L64: It should be clarified that, while using the first 100 years for the regression may be a common approach, the resulting ECS may not be the "true" ECS in the final equilibrium state, especially for models that manifest different slopes for the Gregory plot after the first century (e.g., Winton et al. 2020, JAMES).
- L70: Does the fixed-SST run use the climatological mean SST that repeats itself every year, or does it include the observed SST interannual variabilities?
- L85: It would be helpful to break down the $\xi$ factor into the contributions from the feedback and ERF terms.
- L112–113: This description is somewhat unclear. What does "previous version" refer to, and how is the model "fine-tuned"? And is this sentence describing only the full-sky experiment, or is the clear-sky experiment also spun up with the same initial ocean state from the "previous version"?
- L116–119: As stated in my general comments, it is essential to include a figure for the surface temperature and ice cover of the full-sky and clear-sky experiments.
- L128: It should be clarified that the "different initial conditions" are all taken from the respective control runs after equilibration.

- L142: As stated in my general comments, it should be clarified that the "strong radiative effect" of clouds can affect the other feedback mechanisms by (1) changing the control climate state or (2) changing the thermodynamic/dynamic responses to increased CO2. The physical interpretation is very different between these two scenarios.
- L161–166: As stated in my general comments, it should be clarified (and quantified if possible) whether this damped lapse-rate feedback is due to the lack of cloud-induced upper tropospheric warming, or due to the weaker warming over the tropical convective region that reduces the free-tropical warming globally. To address this question, it may be helpful to compare the 3-dimensional temperature responses between the clear-sky and full-sky cases, and check whether the differences in warming are collocated with the cloud radiative heating (both in height and horizontally), or whether they are more extant below the cloud heating level and beyond the high cloud regions.
- L167–168: The first sentence fits better with the preceding paragraph.
- Figure 2: Please add a column to show the sum. Also, following my comment on L85, it seems that the sum of the clear-sky from full-sky feedback parameters is similar to the sum in the clear-sky experiment. What is the $\xi$ factor and the implied ECS if equation (4) is computed with these "clear-sky from full-sky" values? Is the implied ECS (perhaps coincidentally) similar to the clear-sky values?
- L177: It should be clarified that the CO2 forcing differs because of the cloud masking and the differences in temperature and humidity.
- L178: The hyphen after "temperature" is unnecessary.
- L186–187: The "low bias" here is not simply a problem with denser points away from the y-axis, but it essentially indicates the time-dependent (or temperature-dependent) feedback strength. If the feedback parameter is unchanged over time, all points would sit on the same straight line and no "low bias" would occur despite the denser points away from the y-axis. This comment also applies to L190 where the argument "more homogeneously distributed points reduced the effect of bending" blurs the essential issue of time-dependent feedback.
- L198–199: Why is the enhancement expected? Is it based on the assumption that cloud masking effect dominates over the cloud adjustment? Also, it would be intuitive to refer to Figure 1 here or around L215 for explaining why [G100] differs from [G20], i.e., the clear-sky case shows more "bending" than the full-sky case.
- L214–215: This "artifact" seems physical to me – the southern ocean warming and sea-ice albedo feedback occurs over the centennial timescale, so it is more pronounced in [G100].
- L219–220: The definition of local feedback by global mean temperature has some advantages as the authors describe, but it may blur the physical interpretation of lapse-rate feedback differences between clear-sky and full-sky cases, because the free-tropospheric temperature profiles (and the lapse rate) over most the tropics and subtropics roughly follow the weak temperature gradient mechanism, which is determined by the SST of the convective region rather than the global mean SST. If so, it may be helpful to check whether computing the lapse rate feedback with the tropical-mean SST, or with the precipitation-weighted SST (e.g., Zhang and Fueglistaler 2020, GRL), can help reduce the lapse rate feedback differences.
- Figure 4: Please add a column to show the sum.

- Figure 5: It would be helpful to clarify in the caption that the "total" row is the sum of the "temperature", "water vapor", "albedo", and "cloud" rows, whereas the "temperature" row is approximately the sum of the "planck" and "lapse rate" rows.
- Figure 6: This figure is confusing because the stratospheric adjustments are included in the temperature adjustments. It would be better to show the tropospheric adjustments (planck and lapse rate) rather than the full temperature adjustments, so that they are not washed out by the nearly uniform stratospheric adjustments, and that the "total" row is the sum of all other rows.

---

## Referee Comment (RC2)

Review Mosso et al 2024

This study examined the role of cloud radiative feedbacks by comparing full-sky and clear-sky conditions using two methods. The first method diagnosed the feedback from the difference between full-sky and clear-sky fluxes in the full-sky experiment. The second method involved two experiments: one where the radiative effects of clouds were visible and another where they were transparent.

The authors found discrepancies between the suggested impacts from these two approaches. The diagnosed estimate suggested that clear-sky conditions have more negative feedback, while the experiments indicated that clear-sky conditions have a less negative feedback. The less negative feedback is attributed to the absence of the stabilizing role of clouds.

This study presents a compelling experiment along with intriguing findings. The aim is particularly noteworthy; to quantify the difference in impacts between the immediate impact by removing the clouds, without allowing the system to adjust to this change as in the clear-sky simulation are representative of the masking effect of clouds, and the impacts which are attributable to both simple masking and the feedback mechanisms alteration by global circulation response to cloud transparency.

While the experiment and the result are worth publishing, I have a question regarding the experimental setup designed for this study, specifically aimed at elucidating the role of clouds in radiative feedback.

- The solar insolation is decreased to approximate the control climate to preindustrial conditions. However, there is an asymmetrical treatment regarding the transparency of shortwave and longwave radiation. While the reduced solar insolation implicitly accounts for the preindustrial radiative effect of clouds on net shortwave radiation (i.e., incoming minus outgoing), the radiative effect of clouds on longwave radiation is not considered. This experimental setup may influence feedback mechanisms due to differences in the mean state of the control climate.
  - The greater extent of Southern Ocean sea ice observed in the control experiment, compared to the clear-sky experiment, may be attributed to the enhanced cooling of the sea surface due to reduced downward longwave radiation with transparent clouds in the longwave spectrum. This aspect warrants discussion. For instance, have you conducted a comparative analysis of the budget terms over the sea ice regions between the two experiments?
  - Could you please compare the atmospheric profile in the control climate between the clear-sky and full-sky experiments, particularly focusing on lapse rate and stability? How do these comparisons correlate with the differences in feedbacks observed? It's important to examine and provide explanations for these aspects.
  - Since the solar insolation remains constant between the control and 2xCO2 experiments, the former essentially represents an experiment with fixed shortwave radiative cloud effects. However, the radiative impact of changes in albedo from non-cloud sources is diminished due to the reduced solar insolation. Consequently, comparing radiative feedbacks in units of [Wm-2K-1] may not fully elucidate the mechanisms underlying the differences in feedback. While this point may be relatively minor for feedback over the Southern Ocean sea ice, comparing albedo [0-1] and feedback between full-sky and clear-sky conditions could help clarify differences in surface albedo feedback across different regions.

- I would also appreciate it if the authors could elaborate on the reasons for the discrepancy between the immediate impact of the clear-sky condition and the impact observed after the system's response.

Due to the uncertainties mentioned above, proposing an inter-model comparison to understand the role of cloud feedback using the experimental setup in this manuscript appears premature. Therefore, I recommend removing the sentence that proposes the inter-model comparison.

I add minor comments at the bottom of this document.

My suggestion is for a major revision that addresses the points mentioned above.

Minor comments

Figure 2: Having total feedback in this figure will help readers to see quantitative contributions.

Figure 4: Having total forcing in this figure will help readers to see quantitative contributions.

L72 Define 'CR'.

L158 Refer FAT mechanism by Hartmann and Larson (2002) here.

L158-159 I do not understand this. Explain more explicitly why the non-existence of convergence could weaken the LR feedback.

L229 Why is the difference in forcing in full-sky experiment positive over land and predominantly negative over the oceans?

- An experiment in which any change in clouds are transparent could make more sense.
- Reduce the solar constant to stabilize the equilibrium climate system in the preindustrial climate.
  - Overall, clouds cool the earth
  - The equilibrium state of the clear-sky simulation is colder than full-sky simulation. (-0.61K)
    - Cooler over land
    - Warmer in mid-latitude ocean
    - Increased sea ice over the Southern Ocean
  - Interpret the mechanisms
    - Clouds reduces the radiative cooling from the surface
- ECS high in clear-sky simulation
  - It is difficult to attribute it because of the increase in Forcing. (Fig 3)
- More positive surface albedo feedback in clear-sky simulation
  -

Figure 1. ECS high in clear-sky simulation

Figure 2. Big disagreement between the diagnosed clear-sky feedback with clouds and that from the clear-sky simulation.

- Planck feedback: More negative in clear-sky feedback. The diagnosed value is similar to the simulated value.
- Lapse-rate feedback is similar to full-sky in the diagnosed clear-sky feedback but much less negative in the simulaton.
- Water-vapour feedback is much more positive in the diagnosed clear-sky feedback than the full-sky feedback but less positive in the simulated feedback.
- Albedo feedback: More positive in clear-sky feedback.

---

## Author Comment (AC1)

**Answers to the reviewers for EGUsphere-2024-618- The presence of clouds lowers climate sensitivity in the MPI-ESM1.2 climate model**

We would like to sincerely thank the two anonymous reviewers for their careful and thorough analysis of our manuscript. Their valuable comments and suggestions have been of great help in improving the quality of our manuscript. We acknowledge their concerns and have made significant changes to the manuscript as a result of their input. We reply here by paragraph to the major comments and by point to the minor comments that they raised.

**Responses to Reviewer 1**

*P1:* *This manuscript presents a "clear-sky" modeling approach (i.e., making clouds transparent to radiation) for a more complete evaluation of clouds impacts on the equilibrium climate sensitivity (ECS). In complement to the common approach of diagnosing the cloud radiative effects from the full-sky model experiment (both the cloud direct radiative effects and the cloud masking on CO2 and water vapor radiative effects), this clear-sky approach additionally accounts for the cloud indirect effects on thermodynamics (e.g., temperature lapse rate and water vapor distribution) and dynamics (e.g., changes in circulation and the associated surface warming pattern), which are shown to significantly modulate the radiative forcing, adjustments, and feedbacks. As the result, the ECS in the clear-sky experiment is much higher than the fullsky experiments with the same climate model, which is surprising but interesting. The authors further perform a systematic decomposition of the effective radiative forcing (ERF) and feedbacks and show their spatial patterns, which helps clarify the physical mechanisms.*

*P2:* *Overall, I think this clear-sky approach is useful in identifying the indirect cloud effects on ERF and feedbacks, which have not been emphasized much in previous studies. Unfortunately, the comparison is heavily impacted by the control climate differences between the clear-sky and full-sky run, which complicates the physical interpretation. Although the authors have tuned the global-mean temperature to be comparable by reducing insolation in the clear-sky run, the temperature spatial pattern (pole-to-equator gradient, land-sea contrast, interhemispheric contrast, etc.) is still drastically different, and many of the adjustments and feedbacks are dependent on these patterns (especially the Antarctic sea-ice cover and its associated albedo feedback). Thus, I think it is a bit far-fetched to attribute the clear-sky versus all-sky ERF and feedback differences simply to the cloud modulation of radiative adjustments and feedbacks. In the current manuscript, this complication by control climate has not been clearly presented or sufficiently discussed.*

*P3:* *To address this issue, I think the difference between the control climate states in the clear-sky and all-sky runs should be presented and discussed in further detail. In the current manuscript, it is only mentioned briefly on L116-119 without any figures. I think spatial maps of the top-of-atmospheric (TOA) radiative imbalance and surface temperature for both clear-sky and all-sky runs and their differences should be shown. They are essential for the readers to evaluate the mean-state impacts on ERF and feedbacks (e.g., by comparing them to Figures 5 and 6 ). In addition, the sea surface temperature (SST) warming pattern (evaluated with the same ensemble members and time spans as Figure 5) should also be presented, so that the readers can link the feedback differences to the known mechanisms of SST pattern effects, and possibly associate them with the SST Green's function studies.*
*Reply:* As we now stated more clearly in the introduction, our experimental setup accounts for the role of clouds via control climate SST pattern modification and climate change response together. We significantly extended section 2.4, where we now show both the control temperature pattern difference and the warming patterns normalised by their global mean warming in the full sky and clear sky experiment. The warming patterns are very similar between the two configurations, except for the latitudinal position of the warming peak in the southern ocean, caused by the different sea-ice coverage.

*P4:* *In the result discussions, it would be necessary to distinguish the mean-state induced differences from the cloud-induced differences more clearly. Especially, the impact of the larger Antarctic ice cover in the clear-sky control climate should be more thoroughly analyzed - it not only leads to a stronger ice albedo feedback, but it is also associated with a southern ocean amplified (and tropically damped) SST warming pattern (implied from Figure 5, "planck" panel), which is known for increasing the climate sensitivity (most recently by Kang,*

*Ceppi, Yu and Kang, 2023, Nature Geoscience). Even in clear-sky experiments without cloud feedbacks, the lapse rate feedback is still impacted by the SST pattern (e.g., Andrews and Webb, 2018, Journal of Climate). This mechanism is implied in Figure 5: the SST over the tropical convective regions in the clear-sky run does not warm as much as the full-sky run, which likely leads to the reduced free-tropospheric warming and lapse-rate feedback globally (compare the "planck" and "lapserate" panels). Thus, I think the authors' explanation of the lapse rate feedback differences by the direct cloud radiative heating (L155-160 and L265-266) is likely insufficient.*

**Reply:** To better distinguish between the mean-state and cloud-induced differences, we extended the discussion of the results from the clear-sky fluxes within the full-sky simulations ($CS_F$) to the whole manuscript, which before was only limited to section 3.1. The (small) difference between CS and $CS_F$ is the effect of the global circulation response to the cloud transparency, which includes modifications to the mean state. We also extended the explanation of the mechanisms related to the lapse rate feedback weakening in the clear sky experiment in section 3.1.

**P5:** *Additionally, I think it would be necessary to discuss how this clear-sky approach is compared with the cloud-locking approach (e.g., Medeiros, Clement, Benedict, and Zhang, 2021, npj Climate and Atmospheric Science), which can also disable all cloud-induced thermodynamic and dynamic changes in the warming climate but largely maintains the control climate state. Are there significant weaknesses of the cloud-locking approach that motivate the authors to use the clear-sky approach instead? If so, it would be helpful to discuss in the introduction and conclusion sections. If not, the different physical interpretations of these two approaches should still be discussed. It would be even better if the authors can perform cloud-locking experiments and see which of the cloud impacts on ERF and feedbacks are robust for both approaches.*

**Reply:** The cloud-locking approach has been extensively used to compute the effects of cloud feedbacks on other feedback mechanisms, something that has been referred to as feedback synergy [Mauritsen et al., 2013]. Our setup, differently from previous studies using cloud locking, accounts for the radiative role of clouds as a whole on the response to the $CO_2$ forcing and their dynamical effect on the circulation. Practically, the interpretation is similar to cloud locking, as in our clear-sky setup we prescribe the cloud radiative properties such that their radiative forcing, set to zero, is independent of all other model variables. We think this is relevant when comparing with the *bottom-up* approach used in the analytical framework for the ECS calculation.

**Specific comments**

**C1:** *L13: It should be clarified that the ECS is defined by the global-mean surface temperature (GMST).*
**Reply:** We have clarified it at line 14.

**C2:** *L25: Please include the references for the "recent studies" at the beginning of this line.*
**Reply:** References have been added.

**C3:** *L64: It should be clarified that, while using the first 100 years for the regression may be a common approach, the resulting ECS may not be the "true" ECS in the final equilibrium state, especially for models that manifest different slopes for the Gregory plot after the first century (e.g., Winton et al. 2020, JAMES).*
**Reply:** We clarified it with reference to fig. 1. This is also one of the reasons why we choose to perform CO2 doubling experiment instead of the usual CO2 quadrupling. The smaller the forcing, the smaller the error from linear approximation of N(T) is.

**C4:** *L70: Does the fixed-SST run use the climatological mean SST that repeats itself every year, or does it include the observed SST inter-annual variabilities?*
**Reply:** It contains inter-annual variability. Surface temperatures and sea-ice are prescribed from historical values, which are then doubled in the abrupt2xCO2 experiment.

**C5:** *L85: It would be helpful to break down the $\xi$ factor into the contributions from the feedback and ERF terms.*
**Reply:** The point that we want to make here is that the overall cloud direct effect on the ECS is destabilising as $\xi$ is greater than 1.

***C6:*** *L112-113: This description is somewhat unclear. What does "previous version" refer to, and how is the model "fine-tuned"? And is this sentence describing only the full-sky experiment, or is the clear-sky experiment also spun up with the same initial ocean state from the "previous version"?*

**Reply:** Previous version refers to MPI-ESM1.1. We rephrased the paragraph to better clarify that both clear-sky and full-sky are spun up from the same initial ocean state, and in the clear-sky cloud transparency and solar constant reduction are instantly applied. After that, the clear sky simulation is let to run for another 800 years to allow the system to adjust to those modifications.

***C7:*** *L116-119: As stated in my general comments, it is essential to include a figure for the surface temperature and ice cover of the full-sky and clear-sky experiments.*

**Reply:** It is now included in figure 1 in section 2.4

***C8:*** *L128: It should be clarified that the "different initial conditions" are all taken from the respective control runs after equilibration.*

**Reply:** We clarified it.

***C9:*** *L142: As stated in my general comments, it should be clarified that the "strong radiative effect" of clouds can affect the other feedback mechanisms by (1) changing the control climate state or (2) changing the thermodynamic/dynamic responses to increased $CO_2$. The physical interpretation is very different between these two scenarios.*

**Reply:** In the introduction and at L116-122, we better stated that we are interested in accounting for both of them. Panels e) and f) of Figure 1 also suggest that the effects of a different control climate state are not reflected in an analogously significant difference in the warming pattern.

***C10:*** *L161-166: As stated in my general comments, it should be clarified (and quantified if possible) whether this damped lapse-rate feedback is due to the lack of cloud-induced upper tropospheric warming, or due to the weaker warming over the tropical convective region that reduces the free-tropical warming globally. To address this question, it may be helpful to compare the 3-dimensional temperature responses between the clear-sky and full-sky cases, and check whether the differences in warming are collocated with the cloud radiative heating (both in height and horizontally), or whether they are more extant below the cloud heating level and beyond the high cloud regions.*

**Reply:** We unfortunately do not have 3D cloud radiative heating as a model output. However, we showed that the difference in high altitude heating in the IPWP is confined below the level of maximum cloudiness, suggesting an active role of high clouds for the lapse rate of the region.

***C11:*** *L167-168: The first sentence fits better with the preceding paragraph.*

**Reply:** We adjusted it.

***C12:*** *Figure 2: Please add a column to show the sum. Also, following my comment on L85, it seems that the sum of the clear-sky from full-sky feedback parameters is similar to the sum in the clear-sky experiment. What is the $\xi$ factor and the implied ECS if equation (4) is computed with these "clear-sky from full-sky" values? Is the implied ECS (perhaps coincidentally) similar to the clear-sky values?*

**Reply:** The figure, which is now number 3, has been replotted with the sum. We extended the discussion about $CS_F$, now including the results for ECS, feedback and forcing calculated in this way. The agreement between CS and $CS_F$ is indeed particularly interesting, and we show that for the feedback component, this is because of differences of opposite signs that cancel out.

***C13:*** *L177: It should be clarified that the CO2 forcing differs because of the cloud masking and the differences in temperature and humidity.*

**Reply:** We talk about this when we separate the total ERF in its components with PRP.

***C14:*** *L178: The hyphen after "temperature" is unnecessary.*

**Reply:** It has been removed.

***C15:*** *L186-187: The "low bias" here is not simply a problem with denser points away from the y axis, but it essentially indicates the time-dependent (or temperature-dependent) feedback strength. If the feedback*

*parameter is unchanged over time, all points would sit on the same straight line and no "low bias" would occur despite the denser points away from the y-axis. This comment also applies to* **L190** *where the argument "more homogeneously distributed points reduced the effect of bending" blurs the essential issue of time-dependent feedback.*

**Reply:** We have reworded the sentence to make the link to time-dependent feedback explicit.

**C16:** *L198-199: Why is the enhancement expected? Is it based on the assumption that cloud masking effect dominates over the cloud adjustment? Also, it would be intuitive to refer to Figure 1 here or around L215 for explaining why [G100] differs from [G20], i.e., the clear-sky case shows more "bending" than the full-sky case.*

**Reply:** Cloud masking of $CO_2$ forcing is a strong and homogeneous signal between different models, which has long been known and physically explained [e.g. Stevens and Kluft, 2023]. Cloud adjustment instead is not necessarily shown in all models and is generally of smaller magnitude.

**C17:** *L214-215: This "artifact" seems physical to me - the southern ocean warming and sea-ice albedo feedback occurs over the centennial timescale, so it is more pronounced in [G100].*

**Reply:** The time-scale interpretation makes sense for the sea-ice albedo feedback, but WV and LR show differences of the same magnitude and same sign as their feedback, indicating that feedback gets weaker over time. We generally reworded the sentence in terms of time dependency of the feedback, which can either be intensified or weakened depending on the mechanism.

**C18:** *L219-220: The definition of local feedback by global mean temperature has some advantages as the authors describe, but it may blur the physical interpretation of lapse-rate feedback differences between clear-sky and full-sky cases, because the free-tropospheric temperature profiles (and the lapse rate) over most the tropics and subtropics roughly follow the weak temperature gradient mechanism, which is determined by the SST of the convective region rather than the global mean SST. If so, it may be helpful to check whether computing the lapse rate feedback with the tropical-mean SST, or with the precipitationweighted SST (e.g., Zhang and Fueglistaler 2020, GRL), can help reduce the lapse rate feedback differences.*

**Reply:** From the plot of the normalised warming pattern that we now included, it can be seen that tropical and subtropical areas have near zero normalised warming differences between clear sky and full sky. This suggests lapse-rate feedback differences are not significantly driven by a different SST response at low latitudes. Instead, they can be more precisely linked to the role of high clouds, which we now discuss more extensively in section 3.1.

**C19:** *Figure 4: Please add a column to show the sum.*
**Reply:** A column with the sum has been added.

**C20:** *Figure 5: It would be helpful to clarify in the caption that the "total" row is the sum of the "temperature", "water vapor", "albedo", and "cloud" rows, whereas the "temperature" row is approximately the sum of the "planck" and "lapse rate" rows.*
**Reply:** We made the clarification.

**C21:** *Figure 6: This figure is confusing because the stratospheric adjustments are included in the temperature adjustments. It would be better to show the tropospheric adjustments (planck and lapse rate) rather than the full temperature adjustments, so that they are not washed out by the nearly uniform stratospheric adjustments, and that the "total" row is the sum of all other rows.*
**Reply:** The suggested modifications have been made.

**Responses to Reviewer 2**

*P1: This study examined the role of cloud radiative feedbacks by comparing full-sky and clear-sky conditions using two methods. The first method diagnosed the feedback from the difference between full-sky and clear-sky fluxes in the full-sky experiment. The second method involved two experiments: one where the radiative effects of clouds were visible and another where they were transparent.*

*The authors found discrepancies between the suggested impacts from these two approaches. The diagnosed estimate suggested that clear-sky conditions have more negative feedback, while the experiments indicated that clear-sky conditions have a less negative feedback. The less negative feedback is attributed to the absence of the stabilizing role of clouds. This study presents a compelling experiment along with intriguing findings. The aim is particularly noteworthy; to quantify the difference in impacts between the immediate impact by removing the clouds, without allowing the system to adjust to this change as in the clear-sky simulation are representative of the masking effect of clouds, and the impacts which are attributable to both simple masking and the feedback mechanisms alteration by global circulation response to cloud transparency. While the experiment and the result are worth publishing, I have a question regarding the experimental setup designed for this study, specifically aimed at elucidating the role of clouds in radiative feedback.*

*P2: The solar insolation is decreased to approximate the control climate to preindustrial conditions. However, there is an asymmetrical treatment regarding the transparency of shortwave and longwave radiation. While the reduced solar insolation implicitly accounts for the preindustrial radiative effect of clouds on net shortwave radiation (i.e., incoming minus outgoing), the radiative effect of clouds on longwave radiation is not considered. This experimental setup may influence feedback mechanisms due to differences in the mean state of the control climate.*

**Reply:** The rationale behind the choice of reducing the solar constant is first to compensate for the clouds with a forcing which is sufficiently spatially homogeneous, and secondly to get an experimental setup which can be easily implemented. The process of solar constant tuning does so by balancing both the negative shortwave and positive longwave CRE, which, when combined, result in a negative sum on the global mean but not at the local level. It is also for this reason that we let the system adjust on centennial timescales. When making clouds transparent we are getting rid of both of them. Sophisticated methods to balance out the cloud transparency more closely can certainly be invented, but the idea of our study is also to account for the dynamical role of clouds on the circulation, and their indirect role on non-cloud feedback mechanisms, which manifests also through a different mean state. We find compensating for the cloud transparency in the shortwave a reasonable choice since the majority of the feedbacks that can be affected by the clouds, except for the albedo, are primarily in the longwave.

*P3: The greater extent of Southern Ocean sea ice observed in the control experiment, compared to the clear-sky experiment, may be attributed to the enhanced cooling of the sea surface due to reduced downward longwave radiation with transparent clouds in the longwave spectrum. This aspect warrants discussion. For instance, have you conducted a comparative analysis of the budget terms over the sea ice regions between the two experiments?*

**Reply:** In panel c) of Fig. 1 in section 2.4 we show that the cloud radiative effect is positive over Antarctica, which leads to the strong cooling when removed in the clear-sky experiment. In this context, the role of clouds in limiting the amount of sea ice in the region can be thought of as the biggest indirect effect contributing to their role in reducing the ECS reduction.

*P4: Could you please compare the atmospheric profile in the control climate between the clear-sky and full-sky experiments, particularly focusing on lapse rate and stability? How do these comparisons correlate with the differences in feedbacks observed? It's important to examine and provide explanations for these aspects.*

**Reply:** We now plot the vertical warming profile for the Indian Pacific Warm Pool and show how this is related to the high-cloud cover. This further supports the mechanism of high clouds driving high altitude longwave flux convergence that we now more extensively explain in section 3.1.

*P5: Since the solar insolation remains constant between the control and 2xCO2 experiments, the former essentially represents an experiment with fixed shortwave radiative cloud effects. However, the radiative impact of changes in albedo from non- cloud sources is diminished due to the reduced solar insolation. Consequently, comparing radiative feedbacks in units of [Wm-2K-1] may not fully elucidate the mechanisms underlying the differences in feedback. While this point may be relatively minor for feedback over the Southern Ocean sea ice,*

*comparing albedo [0-1] and feedback between full-sky and clear-sky conditions could help clarify differences in surface albedo feedback across different regions.*

**Reply:** We expect only radiative feedbacks in the shortwave to be impacted by the difference in solar insulation in a linear way, which is one of the reasons why we chose the solar constant reduction in the first place, as it would not affect the longwave feedbacks. If the albedo feedback depends linearly on the solar constant, the normalised albedo feedback in the clear-sky experiment would only be bigger by around 9.6%, which does not change the picture significantly. For this reason we added a sentence about the normalised value at lines 244-246, but we did not modify the panel in Figure 7, keeping the unnormalised spatial distribution of the feedback.

**P6:** *I would also appreciate it if the authors could elaborate on the reasons for the discrepancy between the immediate impact of the clear-sky condition and the impact observed after the system's response.*

**Reply:** We discuss those differences in section 3.1 with a focus on lapse-rate, water-vapour and albedo feedback differences. We extended the explanation of the differences in the lapse rate feedback, including a focus on the Tropical Warm Pool. These differences can now be better appreciated with the extension of the discussion about $CS_F$ (calculation with the clear sky fluxes within the flull sky simulation). If the difference between CS and $CS_F$ is interpreted as a difference due to the system response, its effect is mainly that of a different partitioning of the lapse rate and water vapour feedback, while the total feedback remains essentially the same.

**P7:** *Due to the uncertainties mentioned above, proposing an inter-model comparison to understand the role of cloud feedback using the experimental setup in this manuscript appears premature. Therefore, I recommend removing the sentence that proposes the inter-model comparison.*

**Reply:** We understand that our protocol does not meet the strict requirements for an inter-model comparison project, and we did not formally propose one. However, we aim to encourage other modeling groups to conduct similar experiments, which could provide further insights and perhaps refine future protocols.

*I add minor comments at the bottom of this document.*
*My suggestion is for a major revision that addresses the points mentioned above.*

**Minor comments**

**C1:** *Figure 2: Having total feedback in this figure will help readers to see quantitative contributions.*
**Reply:** The figure, which is now 3, has been updated.

**C2:** *Figure 4: Having total forcing in this figure will help readers to see quantitative contributions.*
**Reply:** Same as for the previous comment.

**C3:** *L72 Define 'CR'.*
**Reply:** We have extended the description of the model configuration, lines 96-98.

**C4:** *L158 Refer FAT mechanism by Hartmann and Larson (2002) here.*
**Reply:** The citation has been included.

**C5:** *L158-159 I do not understand this. Explain more explicitly why the non-existence of convergence could weaken the LR feedback.*
**Reply:** We extended the explanation, with the introduction of an explanatory plot of the vertical warming profile and cloudiness in Tropical Warm Pool.

**C6:** *L229 Why is the difference in forcing in full-sky experiment positive over land and predominantly negative over the oceans?*
**Reply:** The sign of the forcing difference is dominated by the cloud forcing and to a lesser extend by water vapour. We find the question particularly interesting, but it has not been possible to investigate it further with the experimental data we have, as it would require separating the components of the cloud radiative effect from different mechanisms. This feature was already present in a previous version of the model, although it is visible only when calculating the fast adjustment with the Hansen method [e.g. see figure 10 in

Block and Mauritsen, 2013].

**References**

K. Block and T. Mauritsen. Forcing and feedback in the MPI-ESM-LR coupled model under abruptly quadrupled CO2. *Journal of Advances in Modeling Earth Systems*, 5(4):676–691, December 2013. ISSN 19422466. doi: 10.1002/jame.20041. URL http://doi.wiley.com/10.1002/jame.20041.

Thorsten Mauritsen, Rune G. Graversen, Daniel Klocke, Peter L. Langen, Bjorn Stevens, and Lorenzo Tomassini. Climate feedback efficiency and synergy. *Climate Dynamics*, 41(9-10):2539–2554, November 2013. ISSN 0930-7575, 1432-0894. doi: 10.1007/s00382-013-1808-7. URL http://link.springer.com/10.1007/s00382-013-1808-7.

Bjorn Stevens and Lukas Kluft. A colorful look at climate sensitivity. *Atmospheric Chemistry and Physics*, 23(23):14673–14689, November 2023. ISSN 1680-7316. doi: 10.5194/acp-23-14673-2023. URL https://acp.copernicus.org/articles/23/14673/2023/.